# Mobile payment, digital inclusive finance, and residents' consumption behavior research

**Ningning Hu[1]**, **Guanyu Hou [2]** *

**1** Graduate School, Chinese Academy of Fiscal Sciences, Beijing, China, **2** School of Marxism, Northeastern University, Qinhuangdao, China

☯ These authors contributed equally to this work.
* CUPL_hougy21@163.com

**Data Availability Statement:** - Visit the CHFS website (https://chfs.swufe.edu.cn/). - Navigate to the "Data Application" section. - Fill out the application form, indicating the purpose of the data

## Abstract

Based on the data of multiple Chinese household finance surveys, the interactive relationship between mobile payment, inclusive digital finance, and household consumption is discussed. It is found that mobile payment can directly and effectively improve household consumption, and the impact on hedonistic and developmental consumption is greater than survival consumption, which is conducive to upgrading household consumption. At the same time, mobile payment can indirectly promote basic and developmental consumer spending through digital inclusive financial mechanisms and weaken hedonistic consumer spending. The heterogeneity analysis found that the impact of mobile payment on household consumption was affected by income level, dependency structure, and regional attributes, and the low-income and high-income groups benefited more significantly, and the consumption promotion effect in the eastern and central regions was greater than that in the western and northeastern regions. Further research finds that with the improvement of income status, the promotion effect of mobile payment on consumption shows a marginal decreasing trend. It is recommended to continue to promote the popularization and application of mobile payment, accelerate the matching of supply and demand in the consumer market, formulate financial inclusion policies according to local conditions, and form a good interaction mechanism between mobile payment, digital finance, and household consumption.

## Introduction

The rapid development of mobile payment has become one of the important factors driving the growth of the consumer market. With the increasing popularity of mobile payment, consumers' shopping experience has also changed. Whether online or offline, consumers can quickly complete the payment through mobile payment, improving the efficiency and convenience of shopping. At the same time, mobile payment also provides consumers with more shopping choices, such as mobile app stores or major e-commerce platforms, and these platforms can also improve consumers' shopping experience and purchasing power through mobile payment. In addition to convenient and fast payment methods, mobile payment can

usage. - Once approved, the dataset will be provided.

**Funding:** The authors received no specific funding for this work.

**Competing interests:** The authors have declared that no competing interests exist.

also promote the rapid development of the consumer market. Mobile payment can help merchants increase sales and revenue, increase customer loyalty, and provide more business opportunities for emerging e-commerce platforms. Mobile payment can also bring more liquidity to the consumer market, consumers can shop anytime, anywhere, and merchants can also sell anytime, anywhere, which can better meet the needs of consumers, but also provide more impetus for the rapid development of the market.

During the pandemic, the importance of mobile payment has become even more prominent. Mobile payment is more convenient, fast, and secure than traditional cash and bank card payments. With the rapid development of mobile technology and the impact of the epidemic, the mobile payment model has grown rapidly and has become an important channel for everyday consumption, and consumers can complete the payment at any time and place without the need to carry cash or bank cards. In 2020, the global mobile payment users exceeded 1 billion, the market size reached 15 trillion US dollars, and it is expected to exceed 30 trillion US dollars in 2025. Consumers can shop anytime, anywhere, and Alipay transactions exceeded 32.1 billion in 2021, equivalent to 230 per person. Mobile payment plays a key role in the development of e-commerce; for example, in 2021, China's e-commerce transaction scale exceeded 38 trillion yuan, and mobile payment became the main method. The data shows that mobile payment has an important impact on the development of the consumer market, which is convenient and fast, improves merchant sales and revenue, increases consumer convenience, and provides opportunities and financing support for e-commerce platforms and small and micro enterprises.

In recent years, the combination of mobile technology and inclusive finance has made up for the shortcomings of the high cost of traditional finance and limited service population, improved the financial contact mode of micro-subjects, improved the financial resources of vulnerable groups, and also played a positive role in China's consumer market. The post-epidemic era is a critical period for countries around the world to boost consumption and promote the economy, and continuing to promote mobile payment and vigorously developing inclusive digital finance have important theoretical and practical significance for alleviating the current situation of world economic weakness and stimulating the vitality of global markets and consumption.

To this end, this paper uses multiple issues of China Micro Household Survey data to explore the interaction mechanism between mobile payment, inclusive digital finance, and household consumption and attempts to provide some theories and display references for stimulating consumption and boosting the economy. The marginal contributions of this paper are as follows: (1) Using multiple periods of Chinese financial household microdata (CHFS2015-2019), the impact of mobile payment on household consumption level is analyzed. At the same time, this paper refines the highlights of household consumption, compares and analyzes the impact of mobile payment on different consumption categories, and further investigates the upgrading of mobile payment on household consumption structure. (2) Combining mobile payment and inclusive digital finance, explore the role mechanism of inclusive digital finance in mobile payment and household consumption, and analyze the different performance of this mechanism in various types of consumption; (3) Further consider the consumption differences of different income levels, and explore the difference in the impact of mobile payment on consumption with the change of income level. (4) Various instrumental variables were selected from the household, province, and geographical levels for the IV-2sls test, and the robustness test was carried out by combining sub-samples and core explanatory variable substitution to ensure the robustness of the research conclusions.

## Literature review

The literature closely related to the research in this paper has three aspects. The first is research on mobile payment and consumption. Mobile payment has become a popular payment method today, and with the continuous development of mobile technology and the Internet, it has been increasingly widely used in various fields. The system combs through the history of mobile payments and provides various explanations and research methods. The authors believe that adopting mobile payments depends on consumer attitudes and behaviors and multiple factors such as technology, policy, and security [1]. In addition, mobile payment and consumption are hot topics in China's current social and economic development, attracting the attention and research of many scholars. The following are some academic research progress of Chinese scholars in the field of mobile payment and consumption:

(1) Research on user behavior of mobile payment Many scholars study the behavioral characteristics of mobile payment users through questionnaires, experiments, observations, and other methods, including frequency of use, payment amount, payment scenarios, payment methods, etc. For example, some studies have found that female users are more willing to use mobile payments, while older users prefer to use traditional payment methods [2]; At the same time, users who use mobile payment are more inclined to spend on catering, transportation, etc. (2) The impact of mobile payment on consumer behavior.Some scholars have studied the impact of mobile payment on consumer behavior, including consumption frequency, consumption amount, purchase decision, etc [3,4]. For example, some studies have shown that mobile payments can boost consumption [5,6], especially in areas such as e-commerce and catering, and that mobile payments are more attractive to price-sensitive consumers. (3) Security and privacy protection of mobile payment. Since mobile payment involves financial information and personal privacy, some scholars have studied security and privacy protection [7]. Studies have shown that mobile payment has security risks, such as account theft, information leakage, and other issues, so security measures and management need to be strengthened. In addition, some scholars have also studied consumers' attitudes and behaviors toward mobile payment privacy protection, and how to improve consumers' privacy awareness and protection capabilities. (4) The business model and development trend of mobile payment. Some scholars have studied mobile payment's business model and development trend, including the charging method, profit model, and industry competition [8]. For example, some studies point out that the competition in the mobile payment industry will become increasingly fierce, and companies need to strengthen their capabilities in technology research and development, service quality, marketing, and other aspects.

The second is the research on digital inclusive finance and consumption. The digital inclusive finance and consumption field has gradually become the focus of academic research, and many Chinese scholars have devoted themselves to the discussion and research in this field. Professor Chen Xiaoping's research has provided an in-depth and systematic analysis of the development status, characteristics, problems, and challenges of digital financial inclusion in China, providing a valuable contribution to the academic community [3]. In their research, they elaborated on the background of the rapid development of inclusive digital finance in China, such as policy promotion, technological innovation, and social demand, combined with domestic and foreign cases and experiences. They deeply discussed the important role of inclusive digital finance in improving the coverage of financial services, reducing the cost of financial services, and promoting economic development. In addition, some scholars have paid attention to the problems and challenges faced by inclusive digital finance in the process of China's development [9], such as the dilemma of the integration of financial technology and traditional finance, the lack of digital financial risk management and supervision, and the

quality of inclusive financial services, and put forward practical solutions and suggestions for these problems. In these studies, scholars have found that the rapid development of digital financial inclusion in China is due to strong government support and effective regulatory policies. However, this area still needs to overcome a series of challenges, such as high capital costs, fierce market competition, and difficulties in risk management. In response to these problems, scholars believe it is necessary to conduct in-depth research to provide more useful suggestions and solutions for the sustainable development of digital financial inclusion.

Through an academic review of digital financial inclusion and consumption, this field has achieved remarkable development results in China but still needs to be studied in depth on existing issues and challenges [10]. This will help promote the healthy development of digital financial inclusion, create more value for the social economy, and provide a broad research field for most scholars.

The third is research on mobile payment, inclusive digital finance, and consumption.

In "Research on the Impact of Mobile Payment and Digital Inclusive Finance and Consumption," many scholars have conducted in-depth discussions on the relationship between mobile payment, inclusive digital finance, and consumer behavior [11]. Some scholars believe that with the popularization of mobile payment and digital financial inclusion, consumer payment methods have significantly changed, impacting the consumer market [12]. Many empirical studies have found that the development of mobile payment and digital financial inclusion has promoted consumers' willingness to consume and purchasing power [13]. The convenience and security of mobile payments make consumers more willing to make online and offline transactions, thereby increasing overall spending levels. At the same time, inclusive digital finance provides convenient financial services for consumers, lowers the financial threshold, enables more people to obtain credit, insurance, and other financial products, and further promotes consumption growth [14]. In addition, some scholars have also paid attention to the impact of mobile payment and digital financial inclusion on the consumption structure. With the popularity of mobile payment, consumers are more inclined to shop online, and the e-commerce market has developed rapidly. At the same time, digital financial inclusion provides more financing channels for small and micro enterprises and self-employed [15], stimulates innovation and entrepreneurial vitality, and changes the supply structure of the consumer market [16]. While mobile payments and digital financial inclusion have positively impacted the consumer market, academics are also aware of the potential risks [17]. For example, excessive reliance on mobile payments may lead to personal information security issues, while credit approval and risk control in inclusive digital finance still need to be improved [18]. Therefore, authorities and companies must strengthen regulation and technological innovation to ensure that these emerging financial services can continue supporting the consumer market.

## Theoretical hypothesis

Mobile payment includes mobile phone payment, scan code payment, mobile wallet, etc.; its popularity and development can promote residents' consumption expenditure but also affect different levels and fields of consumer expenditure. For survival consumption expenditure, mobile payment can improve the convenience and security of payment. For example, when purchasing daily necessities, using mobile payment can avoid risks such as carrying cash theft, and at the same time, it can be more convenient to purchase daily necessities, thereby increasing the survival security of residents [19].

In terms of commercial service ports, mobile payment can provide more efficient and convenient payment methods, promote the improvement of service supply, and increase consumers' willingness to buy. Through the mobile payment platform, users can book doctors,

purchase educational courses, and order tickets for cultural events and other services online, eliminating the need to wait in line on-site, saving time, and improving the consumer experience. Mobile payment platforms can also provide merchants with more accurate marketing and promotion services, recommend goods and services that meet user needs according to user consumption habits and history, and improve user satisfaction and trust in purchases [20].

Regarding user portals, mobile payment can provide more convenient and diversified consumption methods to increase the user's enjoyment experience. For example, in the fields of tourism, entertainment, and food, users can book air tickets, hotels, attraction tickets, catering, and other services through mobile payment platforms, which can not only enjoy more discounts and benefits but also conveniently complete the entire consumption process and improve the consumption experience. In addition, the mobile payment platform can also provide users with diversified payment methods, such as scan code payment, fingerprint recognition payment, voice payment, etc., to increase users' consumption choices and improve users' payment experience and security.

All in all, mobile payment has a positive effect on the development of consumer expenditure and the increase of residents' development opportunities and hedonistic experience, which can provide merchants with more efficient and accurate marketing and promotion services, and provide users with more convenient and diverse consumption and payment methods. However, the role of mobile payment in promoting specific consumer segments is different. For example, in the catering field, mobile payment can provide a more convenient way to order and checkout, promoting an increase in consumption; In the field of tourism, mobile payment can provide more convenient services such as attraction tickets and hotel reservations, thereby promoting the increase of tourism consumption [21]. Nevertheless, in some areas, such as real estate, the role of mobile payments is relatively limited. Given this, we propose the following hypothesis:

H1: Mobile payment can effectively promote household consumption expenditure, and there are differences in the promotion effect in specific consumption types.

With the continuous development of mobile payment technology, more and more people are beginning to adopt mobile phones or other mobile devices for payment. Mobile payment is more convenient, safe, and fast than traditional cash or bank card payment methods. The widespread popularity of mobile payment brings great convenience to individuals and many benefits to the country's economic development, which is to help reduce the Engel coefficient of residents and promote the upgrading of consumption levels. Generally speaking, countries and regions with low Engel coefficients have relatively high living standards and the spending power of their residents. The wide application of mobile payment is conducive to reducing the transaction cost of goods and services, reducing the cost of living for residents, and helping reduce the Engel coefficient of residents [22].

At the same time, mobile payments can also help drive up consumption levels. Traditional payment methods have limitations, such as people needing to carry cash or bank cards. However, in some cases, these payment methods may not meet the needs of consumers. In contrast, mobile payment provides more payment methods, such as QR code payment, NFC payment, fingerprint payment, etc., which are more convenient and can meet consumers' personalized payment needs. These payment methods can also provide more consumption data and consumption scenarios, help merchants better understand the needs of consumers, provide more personalized services and products, and promote the improvement of consumption levels. Given this, we propose the following hypothesis:

H2: Mobile payment can help reduce the Engel coefficient of residents and help upgrade consumption levels.

Regarding the integration of mobile payment and inclusive digital finance, the role of the rise and development of mobile payment in digital finance must be addressed. Since the mobile payment user base is more likely to benefit from the convenience and services provided by digital financial inclusion, mobile payment is painless, convenient, and low-cost, effectively promoting consumption. The development of digital financial inclusion has also become one of the important factors affecting consumer behavior, and its digital nature means that it is highly dependent on digital technology and mobile networks [17]. As a key link connecting digital financial inclusion, mobile payment is also important to connect users' consumption and financial resource services. Therefore, the widespread application of mobile payment will strongly promote the development of digital financial inclusion.

Through mobile payment, household consumption expenditure has been effectively promoted and improved. First, mobile payments provide consumers a more convenient and secure way to pay, reducing the risk and inconvenience of using cash. Secondly, mobile payment can also provide more flexible payment methods, such as installment payments, coupons, etc., which help promote consumers' desire and spending levels [19]. In addition, through mobile payment, consumers can better grasp their consumption and keep abreast of their financial situation to make more rational consumption decisions. Given this, we propose the following hypothesis:

H3: Mobile payment can promote household consumption through digital financial inclusion.

With the popularity of mobile payment and the development of digital finance, consumers' consumption behavior has been more affected. The convenience and speed of mobile payments make it easier for consumers to make hedonistic purchases, such as online shopping, ordering food, and traveling. Mobile payments can also incentivize consumers to spend more by providing coupons, points, etc. On the other hand, the development and popularization of digital financial inclusion have provided consumers with more consumption choices and support. Through loans and credit services provided by digital finance, consumers can more easily achieve subsistence consumption, such as buying a home and medical services [20]. Digital finance can also promote development-oriented consumption, such as providing loans for education and training and helping consumers improve their quality and skills, employability, and income levels.

Audiences for different types of consumption also differ under the influence of mobile payment and digital finance. For example, young people are more likely to make purchases through mobile payments, while older people are more inclined to use traditional payment methods. For subsistence and development-oriented consumption, low-income households and small and medium-sized enterprises need the support of digital financial inclusion. Therefore, mobile payment and digital financial inclusion are non-negligible in influencing consumer behavior [18]. Consumers can make reasonable use of these tools according to their own needs and circumstances to achieve more rational and effective consumption behavior. At the same time, governments and enterprises should also actively promote the development of digital financial inclusion and provide more support and assistance to consumers.

H4: The transmission effect of mobile payment on consumption through digital financial inclusion will vary according to consumption category.

The widespread application of mobile payment has a significant effect on promoting consumption. However, the heterogeneity of this promotion effect is affected by many factors.

First of all, consumers' economic income level is an important factor affecting their use of mobile payment for consumption. Compared with cash, mobile payment requires users to have a certain economic strength to support their consumption behavior. As a result, people with higher income levels are more likely to use mobile payments for consumption. At the same time, the consumption demand for goods and services of high-income groups is also broader and more diversified, so mobile payment has a more significant impact on the consumption expenditure of high-income groups. Secondly, regional attributes will also affect the promotion of mobile payment. In some developed regions, mobile payment has become widely used and habitualized among consumers, and consumers in such regions are more likely to accept mobile payments and make purchases. In some less developed regions, consumers' consumption behavior is more dependent on cash payment, and the penetration rate of mobile payment is relatively low. Therefore, in less developed regions, the promotion effect of mobile payment is relatively small. Finally, family support pressure will also have a certain impact on the promotion of mobile payments [17]. Consumers with greater family support pressure pay more attention to daily expenses, and mobile payment will impact their consumption decisions. Therefore, consumers with less pressure to use mobile payment are more likely to use it for consumption than consumers with more pressure to raise them. Given this, we propose the following hypothesis:

H5: The promotion effect of mobile payment on household consumption will be heterogeneous due to different economic income levels, regional attributes, and family support pressure.

## Data sources, variable selection, and indicator construction

### Data sources

The data sources used in this article include three sources. First, the micro-level household data comes from the China Household Finance Survey (CHFS) project conducted nationwide by the Southwestern University of Finance and Economics in 2015, 2017, and 2019. The survey is conducted every two years, covering 29 provinces (autonomous regions and municipalities directly under the central government), 355 districts and counties, and 1428 communities (villages), collecting detailed information on household demographic characteristics, assets and liabilities, insurance and protection, expenditure and income, financial knowledge, subjective attitudes, and other aspects. Second, the data at the regional level are mainly from the data published by the China Statistical Yearbook, the official website of the National Bureau of Statistics, and the websites of provincial and municipal statistical bureaus. Finally, the digital financial inclusion data comes from the index compiled by the research team of the Digital Finance Research Center of Peking University and the Ant Group Research Institute. The index covers 31 provinces (autonomous regions and municipalities directly under the central government), 337 cities above the prefecture level (regions, autonomous prefectures, leagues, etc.), and about 2,800 counties (county-level cities, banners, municipal districts, etc.).

### Variable selection and statistical description

1. The variable being explained

This paper uses household consumption expenditure as a measure of consumption level. Drawing on the classification of household consumption by Huang Mengqi and Kim Jong-fan (2022), the household consumption category is subdivided into basic, developmental, and hedonistic consumption expenditure. Basic types include food, housekeeping, property,

clothing, decoration, and daily necessities expenses; Development includes education, health-care, transportation, and network spending; Hedonic types include entertainment, tourism, cars and motorcycles, and other consumer expenditures.

2. Explanatory variables

There are two core explanatory variables in this paper: mobile payment and whether to use mobile phones as proxy variables for mobile payment in the questionnaire taken, of which smartphone assignment is 2, mobile phone assignment is 1, and no mobile phone assignment is 0.

3. Mediation variables

The variable selected for the transmission mechanism of this paper is the digital financial inclusion index, which is based on the research of Yin Zhichao et al. (2021), and the digital inclusive financial index at the district and county levels in 2014, 2016, and 2018 is matched with the CHFS data in 2015, 2017 and 2019. In order to eliminate the influence of outliers on the estimation results, this paper performs logarithmic processing of the digital financial inclu-sion index according to the practice of Fu Qiuzi and Huang Yiping (2018).

4. Control variables

Combined with the existing literature, this paper mainly selects the control variables from three aspects: first, the individual characteristics of the respondents, mainly including the respondents' age, gender, education level, physical condition, nature of household registration, and participation in commercial insurance; Second, the family characteristics of the respon-dents, mainly including total household assets, family size, risk appetite, financial literacy, total family income, family child dependency ratio, old-age dependency ratio, etc.; The third is the regional characteristics of the interviewed households, including urban development level, per capita disposable income, consumer price index, etc.

5. Data Processing

In this study, to obtain more accurate and reliable results, we removed missing values and retained complete household data for three consecutive periods, resulting in 13236 samples. These samples covered 4412 households in 29 provinces and regions in China and followed strict criteria in variable selection. The specific variable descriptions are shown in Tables 1 and 2. Table 1 lists consumption-related variables for sample households, reflecting the consump-tion characteristics of households. As can be seen from the table, there is a clear difference in the quantity of household consumption, with a maximum value of 895,644 yuan and a mini-mum of 180 yuan. Regarding consumption structure, basic consumption still dominates, higher than the level of developmental and hedonistic consumption. In addition, food con-sumption expenditure is the highest, indicating that the Engel coefficient of Chinese house-holds is higher. It is worth noting that transportation, education, and property consumption show large differences in deviation. Table 2 shows basic information on other relevant vari-ables, including household head characteristics, regional characteristics, economic characteris-tics, dependency status, and region-level related variables. The combined consideration of these variables will help us better understand the influencing factors and characteristics of household consumption.

## Model settings

When analyzing the impact of digital financial inclusion development on household consump-tion level, the following benchmark regression model is constructed to analyze the relationship

**Table 1. Descriptive analysis of consumption-related variables.**

| Variables | Obs | Mean | Std. Dev. | Min | Max | p1 | p99 | Skew. |
|---|---|---|---|---|---|---|---|---|
| Total consumption expenditure | 13236 | 43421.937 | 49554.828 | 180 | 895644 | 2880 | 236200 | 5.025 |
| Basic consumption | 13236 | 27145.233 | 32153.126 | 180 | 859720 | 1710 | 140600 | 7.07 |
| Development-oriented consumption | 13236 | 13379.466 | 26660.428 | 0 | 800600 | 120 | 107469 | 10.467 |
| Hedonic consumption | 13236 | 2897.464 | 13291.146 | 0 | 531100 | 0 | 42199.996 | 15.35 |
| Food consumption | 13236 | 17100.152 | 16002.539 | 0 | 432000 | 660 | 64800 | 5.232 |
| Property consumption | 13236 | 3084.545 | 5850.347 | 0 | 240000 | 72 | 23999.999 | 16.546 |
| Transportation consumption | 13236 | 2279.956 | 11772.928 | 0 | 600000 | 0 | 24000 | 27.018 |
| Online consumption | 13236 | 1684.958 | 2536.89 | 0 | 72000 | 0 | 9600 | 10.342 |
| Clothing consumption | 13236 | 1869.316 | 3772.41 | 0 | 160000 | 0 | 15000 | 13.539 |
| Entertainment consumption | 13236 | 612.388 | 2468.55 | 0 | 60000 | 0 | 10000 | 12.404 |
| Health care consumption | 13236 | 6615.828 | 20738.845 | 0 | 800000 | 0 | 79999.961 | 13.938 |
| Tourism consumption | 13236 | 1322.185 | 5944.331 | 0 | 300000 | 0 | 24999.994 | 15.611 |
| Education consumption | 13236 | 2798.719 | 9445.569 | 0 | 400000 | 0 | 35000 | 16.711 |
| Daily consumption | 13236 | 1854.389 | 3686.419 | 0 | 120000 | 0 | 12000.001 | 10.593 |

between digital financial inclusion and household consumption:

$$Comsume_{i,t} = \alpha_0 + \alpha_1 DIF_{i,t} + \sum_{\lambda} \beta_{\lambda} Conrtol_{\lambda,i,t} + u_{i,t} + \varepsilon_{i,t}$$

**Table 2. Descriptive analysis of model variables.**

| Variables | Obs | Mean | Std. Dev. | Min | Max | p1 | p99 | Skew. |
|---|---|---|---|---|---|---|---|---|
| Total consumption expenditure | 13236 | 10.262 | .931 | 5.193 | 13.705 | 7.966 | 12.372 | -.201 |
| Basic consumption | 13236 | 9.806 | .918 | 5.193 | 13.664 | 7.444 | 11.854 | -.266 |
| Development-oriented consumption | 13236 | 8.601 | 1.525 | 0 | 13.593 | 4.787 | 11.585 | -1.197 |
| Hedonic consumption | 13236 | 4.045 | 3.687 | 0 | 13.183 | 0 | 10.65 | .047 |
| Mobile payments | 13236 | 1.459 | .584 | 0 | 2 | 0 | 2 | -.54 |
| age | 13236 | 58.403 | 12.492 | 17 | 95 | 30 | 85 | -.155 |
| The square of age | 13236 | 3566.9 | 1448.211 | 289 | 9025 | 900 | 7225 | .347 |
| gender | 13236 | .785 | .411 | 0 | 1 | 0 | 1 | -1.39 |
| Level of education | 13236 | 8.676 | 4.044 | 0 | 20 | 0 | 16 | -.125 |
| Degree of health | 13236 | .791 | .406 | 0 | 1 | 0 | 1 | -1.435 |
| Nature of account | 13236 | .636 | .481 | 0 | 1 | 0 | 1 | -.564 |
| Family size | 13236 | 3.098 | 1.611 | 1 | 19 | 1 | 8 | 1.302 |
| Child dependency ratio | 13236 | .137 | .288 | 0 | 4 | 0 | 1 | 2.979 |
| Old age dependency ratio | 13236 | .177 | .399 | 0 | 3 | 0 | 2 | 2.744 |
| Risk appetite | 13236 | .227 | .419 | 0 | 1 | 0 | 1 | 1.302 |
| Financial literacy | 13236 | 0 | .553 | -.807 | 1.269 | -.807 | 1.109 | .232 |
| Whether or not to engage in business | 13236 | .116 | .321 | 0 | 1 | 0 | 1 | 2.392 |
| Participate in commercial insurance | 13236 | .08 | .272 | 0 | 1 | 0 | 1 | 3.089 |
| Whether the household is in debt or not | 13236 | .374 | .484 | 0 | 1 | 0 | 1 | .52 |
| Total household assets | 13236 | 12.475 | 1.679 | 0 | 17.845 | 8.154 | 15.897 | -.552 |
| Household income level | 13236 | 10.093 | 2.17 | -1.286 | 15.878 | 0 | 13.017 | -2.809 |
| The level of urban development | 13236 | 58.181 | 11.913 | 36.3 | 89.3 | 38.47 | 89.3 | .833 |
| Per capita disposable income | 13236 | 8.035 | 4.067 | 2.061 | 11.851 | 2.087 | 11.851 | -.687 |
| Consumer Price Index | 13236 | 101.947 | .377 | 101.1 | 103.2 | 101.1 | 103.2 | .432 |

*Consume$_{i,t}$* Represents household consumption in period t; *DIF$_{i,t}$* represents the core explanatory variable of this paper: the Financial Inclusion Development Index; $\sum_\lambda \beta_\lambda Control_{\lambda,i,t}$ It represents the control variables in the model, including the control variables at four levels: household head characteristics, household size and population structure, household property, and regional characteristics; $u_{i,t}, \varepsilon_{i,t}$ Time-fixed effects and random perturbation terms representing the ith family in the t period.

## Empirical analysis

### Benchmark regression analysis

1. Regression analysis of mobile payment on household consumption level

The basic regression results are shown in Table 3, in which column (1) only considers the core explanatory variable mobile payment, and the explanatory variable, consumer expenditure, and the regression results show that the regression coefficient of mobile payment on consumption is 0.66, which is significantly positive at the 1% level. Columns (2) to (5) show the regression results of the model gradually increasing the household head characteristic variable, characteristic economic variable, regional characteristic variable, and time variable, and the results show that although the coefficient of mobile payment has changed, it remains positive, significant at the 1% level and the fitting R2 increases to 46.54%. The regression results of columns (1)-(5) indicate that mobile payment can effectively improve household consumption levels.

From the perspective of the characteristic variables of household heads, the inverted "U" relationship between age and household consumption level shows an inverted "U" shape, indicating that with the increase of age, the household consumption level shows a positive growth trend. However, there will be a downward trend after reaching a certain level. Improving the human capital level of the family can effectively promote the increase of consumption level, and the female head of household has a higher role in promoting household consumption than the male head of household. From the perspective of household population structure, the larger the family size, the more obvious the promotion effect on consumption; The pressure of child support will reduce the consumption expenditure of families, while the impact of the old-age dependency ratio on consumption is not obvious. In addition, household assets and income levels also positively affect consumption. At the same time, household insurance behavior, business behavior, risk appetite, and financial literacy promote household consumption expenditure. The urbanization rate, economic level, and consumer price index of regional control variables showed a positive correlation with household consumption.

2. Regression analysis of mobile payment on different consumption categories of households

First, this paper divides consumption into three categories: survival, development, and hedonic consumption, and conducts regression analysis of these three types of consumption. Specifically, we used the model shown in columns (1)-(3) in Table 4. The results show that mobile payment can effectively improve the level of these three types of consumption, among which the promotion effect on hedonistic consumption is the most significant, followed by developmental and survival consumption. This is because after meeting the basic survival needs, the rapid development and popularization of mobile payment have made it easier for families to access the hedonistic services provided by the supply side and can consume more conveniently, thereby gradually increasing the family's demand for life safety, physical health, and spiritual enjoyment. This conclusion is consistent with the regression results in Table 3. Secondly, this paper refers to existing research, defines the proportion of developmental and

**Table 3. Baseline regression results (I).**

| | (1) | (2) | (3) | (4) | (5) |
|---|---|---|---|---|---|
| | Household consumption level | | | | |
| Mobile payments | 0.6616*** | 0.3269*** | 0.2218*** | 0.2042*** | 0.1557*** |
| | (0.0126) | (0.0139) | (0.0133) | (0.0131) | (0.0135) |
| age | | -0.0280*** | -0.0304*** | -0.0301*** | -0.0305*** |
| | | (0.0040) | (0.0038) | (0.0037) | (0.0037) |
| The square of age | | 0.0002*** | 0.0002*** | 0.0002*** | 0.0002*** |
| | | (0.0000) | (0.0000) | (0.0000) | (0.0000) |
| gender | | -0.0642*** | -0.0459*** | -0.0380** | -0.0339** |
| | | (0.0166) | (0.0156) | (0.0155) | (0.0154) |
| educate | | 0.0299*** | 0.0142*** | 0.0133*** | 0.0145*** |
| | | (0.0021) | (0.0020) | (0.0020) | (0.0020) |
| Degree of health | | 0.0076 | -0.0672*** | -0.0708*** | -0.0630*** |
| | | (0.0166) | (0.0157) | (0.0155) | (0.0154) |
| Nature of account | | -0.5109*** | -0.3348*** | -0.3250*** | -0.3498*** |
| | | (0.0163) | (0.0159) | (0.0156) | (0.0156) |
| Family size | | 0.1400*** | 0.1065*** | 0.1221*** | 0.1302*** |
| | | (0.0049) | (0.0047) | (0.0047) | (0.0047) |
| Child dependency ratio | | -0.0422 | -0.0037 | -0.0195 | -0.0426* |
| | | (0.0265) | (0.0250) | (0.0246) | (0.0244) |
| Old age dependency ratio | | 0.0194 | 0.0266* | 0.0269* | 0.0232 |
| | | (0.0168) | (0.0158) | (0.0156) | (0.0154) |
| Whether or not to engage in business | | 0.2533*** | 0.1353*** | 0.1560*** | 0.1532*** |
| | | (0.0209) | (0.0199) | (0.0196) | (0.0195) |
| Participate in commercial insurance | | 0.1819*** | 0.1066*** | 0.0988*** | 0.0975*** |
| | | (0.0246) | (0.0232) | (0.0228) | (0.0226) |
| Whether the household is in debt or not | | 0.0827*** | 0.1178*** | 0.0839*** | 0.0777*** |
| | | (0.0137) | (0.0129) | (0.0132) | (0.0131) |
| Risk appetite | | -0.0500*** | -0.0644*** | 0.0386** | 0.0656*** |
| | | (0.0166) | (0.0156) | (0.0168) | (0.0168) |
| Financial literacy | | 0.1794*** | 0.1029*** | 0.0914*** | 0.0910*** |
| | | (0.0141) | (0.0134) | (0.0132) | (0.0131) |
| Total household assets | | | 0.1651*** | 0.1466*** | 0.1455*** |
| | | | (0.0047) | (0.0048) | (0.0047) |
| Household income level | | | 0.0472*** | 0.0421*** | 0.0409*** |
| | | | (0.0031) | (0.0030) | (0.0030) |
| The level of urban development | | | | 0.0087*** | 0.0038*** |
| | | | | (0.0006) | (0.0008) |
| Per capita disposable income | | | | -0.0269*** | 0.1316*** |
| | | | | (0.0017) | (0.0289) |
| Consumer Price Index | | | | 0.0730*** | 0.0697*** |
| | | | | (0.0171) | (0.0171) |
| year | NO | NO | NO | NO | Yes |
| Constant terms | 9.2972*** | 10.1774*** | 8.0935*** | 0.6284 | -0.4734 |
| | (0.0198) | (0.1236) | (0.1266) | (1.7345) | (1.8007) |
| Observations | 13236 | 13236 | 13236 | 13236 | 13236 |
| $R^2$ | 0.1721 | 0.3636 | 0.4372 | 0.4561 | 0.4654 |

Note: ***, **, * indicate significant at the level of 1%, 5%, and 10%, respectively, and the standard error in parentheses.

**Table 4. Baseline regression results (II).**

|  | (1) | (2) | (3) | (4) | (5) |
|---|---|---|---|---|---|
|  | **Survival consumption** | **Development-oriented consumption** | **Hedonic consumption** | **Consumption upgrades** | **Consumption downgrade** |
| Mobile payments | 0.1441*** | 0.3246*** | 0.5481*** | 0.0110*** | -0.0119*** |
|  | (0.0135) | (0.0253) | (0.0603) | (0.0040) | (0.0040) |
| Control variables | YES | YES | YES | YES | YES |
| Constant terms | -6.4471*** | 19.2828*** | -46.7490*** | 2.5831*** | -2.3192*** |
|  | (1.7976) | (3.3821) | (8.0624) | (0.5073) | (0.4988) |
| $N$ | 13236 | 13236 | 13236 | 13236 | 13236 |
| $R^2$ | 0.4519 | 0.2969 | 0.3161 | 0.0687 | 0.0771 |
| F | 497.0790 | 255.0194 | 279.0701 | 49.8352 | 56.3060 |

Note: ***, **, * indicate significant at the level of 1%, 5%, and 10%, respectively, and the standard error in parentheses.

hedonistic consumption as the consumption structure upgrade index, defines the proportion of food expenditure as the consumption downgrade index, and discusses the impact of mobile payment on consumption by analyzing the upgrading and reduction of consumption. Specifically, we used columns (4)-(5) in Table 4 to present the findings. The results show that mobile payment has played a positive role in promoting the upgrading of household consumption structure and hurts the consumption downgrade of households. This once again confirms the positive effect of mobile payment on household consumption.

3. On the return of mobile payment to household segmented consumption

We have carefully classified the above three consumption types, including food, clothing, daily necessities, housekeeping property, transportation costs, network expenses, entertainment, health care, travel, and education consumption expenditure, etc., and the regression results for these categories are shown in Table 5. The results show that mobile payment helps to promote various household consumption expenditures and has a significant positive impact on network expenses, clothing expenditures, entertainment expenses, and other aspects. It is worth noting that mobile payment and family education expenditure shows a negative relationship. This may be due to the popularity of mobile networks, the convenience of online education, the spillover effect of knowledge, and the rapid development of online education, which has brought a certain degree of impact to traditional offline education expenditure, resulting in a negative relationship between mobile payment and education expenditure.

## Tool variables (IV-2SLS)

In order to ensure the robustness of the conclusions of this study, this study adopts the instrumental variable method for regression analysis to avoid endogenous problems caused by

**Table 5. Baseline regression results (III).**

|  | **Food expenditure** | **Property expenses** | **Transportation expenses** | **Network spending** | **Clothing expenses** |
|---|---|---|---|---|---|
| Mobile payments | 0.1280*** | 0.1557*** | 0.4280*** | 0.7195*** | 0.4576*** |
|  | (0.0172) | (0.0201) | (0.0623) | (0.0259) | (0.0479) |
|  | **Entertainment spending** | **Medicare spending** | **Tourism expenditure** | **Spending on education** | **Expenditure on daily necessities** |
| Mobile payments | 0.2624*** | 0.1506** | 0.2864*** | -0.1217* | 0.1864*** |
|  | (0.0523) | (0.0624) | (0.0541) | (0.0630) | (0.0307) |

Note: ***, **, * indicate significant at the level of 1%, 5%, and 10%, respectively, and the standard error in parentheses.

**Table 6. Tool variable regression (IV-2SLS).**

| | Family level | Provincial level | Geographical level | |
|---|---|---|---|---|
| | Whether there is a network equipment spend | Provincial smartphone ownership | Spatial distance | Straight-line distance |
| Mobile payments | 0.6980*** | 0.1660*** | 0.8930*** | 0.8831*** |
| | (0.0774) | (0.0151) | (0.1107) | (0.1163) |
| Control variables | YES | YES | YES | YES |
| Constant terms | -3.5462* | -3.0161* | 1.0529 | 1.0112 |
| | (1.9256) | (1.8167) | (1.9920) | (1.9932) |
| N | 13236 | 13236 | 12723 | 12723 |
| R² | 0.3992 | 0.4645 | 0.3416 | 0.3448 |
| F value | 498.92 | 4529.42 | 426.56 | 424.69 |

Note: ***, **, * indicate significant at the level of 1%, 5%, and 10%, respectively, and the standard error in parentheses.

missing variables or reverse causality in benchmark regression. In terms of the selection of instrumental variables, this study considers three aspects: first, from the household level, whether households have network equipment expenditure as a proxy variable to replace mobile payment because network equipment expenditure has a close correlation with mobile payment, but does not directly affect household consumption expenditure; Secondly, from the provincial level, the average smartphone ownership rate of the province where the household is located is used as the proxy variable of mobile household payment, because the provincial smartphone penetration rate is correlated and exogenous with the instrumental variable. Finally, from the geographical level, considering the origin and development of mobile payment in Hangzhou in China, this study selects the straight-line distance or spherical distance between the provincial capital city where the family is located and Hangzhou as the geographical tool variable because this variable has a certain correlation with the use of the mobile household payment. However, the direct impact on household consumption behavior is small, and the correlation and exogenous requirements of the tool variable are met. In this study, the least squares method (2SLS) was used for regression, and the regression results are shown in Table 6. The results show that the impact of mobile payment on household consumption is still significant after alleviating the endogenous problem, which proves the robustness of the research conclusions.

## Robustness test

In order to better verify the validity and robustness of the model, we use two methods for robustness testing. First, we use the sample substitution method to conduct a regression analysis of the impact of mobile payment on household consumption. We used a staging sample from 2015, 2017, and 2019 to conduct regression tests, and the results showed that mobile payments could effectively promote household consumption regardless of the period. Secondly, we use the core explanatory variable substitution method to verify whether mobile payment's impact on household consumption is robust. Specifically, we replaced the online consumption metric with the mobile payment metric and re-conducted the regression analysis. The results show that the positive effect of mobile payment on household consumption is still significant after replacing the core explanatory variables, which indicates that the promotion effect of mobile payment on household consumption is robust. In addition, we constructed a mobile payment factor score, taking into account indicators such as smartphone, non-smartphone, online shopping behavior, the proportion of online shopping, and investment in network equipment, and analyzed them as proxy variables for mobile household payment. After

**Table 7. Robustness test.**

| | Robustness Test (1) | | | Robustness Test (2) | |
|---|---|---|---|---|---|
| | 2015 | 2017 | 2019 | Replace the core explanatory variables | |
| Mobile payments0 | 0.1524*** | 0.1580*** | 0.1436*** | | |
| | (0.0243) | (0.0222) | (0.0229) | | |
| Mobile payments1 | | | | 0.1149*** | |
| | | | | (0.0231) | |
| Mobile payments2 | | | | | 0.1766*** |
| | | | | | (0.0122) |
| Year | NO | NO | NO | YES | YES |
| Control variables | YES | YES | YES | YES | YES |
| Constant terms | -13.8105*** | 4.9269** | 2.9862 | -0.5330 | -1.1824 |
| | (3.6546) | (2.4663) | (3.4034) | (1.8095) | (1.7968) |
| N | 4412 | 4412 | 4412 | 13236 | 13236 |
| R$^2$ | 0.4749 | 0.4537 | 0.4629 | 0.4610 | 0.4684 |
| F | 200.5001 | 193.7848 | 191.0767 | 515.4571 | 530.9724 |

Note: ***, **, * indicate significant at the level of 1%, 5%, and 10%, respectively, and the standard error in parentheses.

replacing the total factor score, the results show that mobile payment still positively impacts household consumption levels, which once again proves the robustness of the conclusion that mobile payment can improve household consumption levels (in Table 7).

## Mechanism analysis

### Test for conduction mechanisms

1. Mechanism test on mobile payment, inclusive digital finance and household consumption

The rapid development of mobile payment can effectively promote household consumption levels. Combined with existing research and literature theoretical analysis, the development of inclusive digital finance has improved household users' economic environment, promoted consumer market development, and increased household consumption to a certain extent. Whether the promotion and use of mobile payment can impact consumption through inclusive digital finance is exactly the transmission path discussed in this article. Therefore, drawing on the research methods of the conduction mechanism in existing studies, the mediation model and the Sobel test are adopted as the mechanism analysis methods, and the mechanism analysis results are shown in Table 8.

The rapid development of mobile payment can effectively promote household consumption. Combined with the existing research and literature theory analysis, the development of inclusive digital finance has improved the economic environment of household users, promoted the development of the consumer market, and improved household consumption to a certain extent. To further explore the influence mechanism of mobile payment on consumption in the path of digital financial inclusion, the mechanism analysis is carried out using the intermediary model and Sobel test, and the analysis results are shown in Table 8. It can be seen that the regression coefficient of mobile payment has a significant positive relationship, and the influence of inclusive digital finance in the return of mobile payment is not significant. However, the third step, complete model regression, and Sobel test results show that mobile payment can improve household consumption through inclusive digital finance. In addition, the Sobel test also shows that the intermediary effect of mobile payment on household consumption through digital financial inclusion accounts for 25.25% of the total effect.

**Table 8. Conduction mechanism test (1).**

| | Intermediary inspection | | | Sobel test |
|---|---|---|---|---|
| | Household consumption | Mobile payments | Household consumption | Household consumption |
| Mobile payments | 0.1633*** | | 0.1628*** | 0.1584*** |
| | (0.0135) | | (0.0134) | (0.0134) |
| Digital financial inclusion | | 0.0331 | 0.3967*** | 0.2347*** |
| | | (0.0473) | (0.0732) | (0.0153) |
| Control variables | YES | YES | YES | YES |
| Constant terms | -0.1835 | 2.2703* | -0.9046 | 1.7534 |
| | (1.8034) | (1.1688) | (1.8064) | (1.7231) |
| N | 13236 | 13236 | 13236 | 13236 |
| R² | 0.4634 | 0.4301 | 0.4646 | 0.4637 |
| Sobel test | | ———— | | 25.25% |
| F | 545.3343 | 476.6055 | 523.0008 | 573.0912 |

Note: ***, **, * indicate significant at the level of 1%, 5%, and 10%, respectively, and the standard error in parentheses.

2. Mechanism test on mobile payment, inclusive digital finance, and consumption upgrading

Next, we will continue to explore whether inclusive digital finance acts as a transmission path between mobile payment and household consumption in the three major consumption areas and consumption upgrades, using the level of survival-oriented consumption, developmental consumption, hedonistic consumption, and consumption upgrading and downgrading as explanatory variables, and the analysis results are shown in Table 9. Studies have shown that mobile payment can affect household consumption through digital financial inclusion, confirming that digital financial inclusion's transmission path exists. Specifically, mobile payment promotes survival-oriented consumption, development-oriented consumption, and consumption upgrading through inclusive digital finance, with intermediary effects of 18.79%, 30.12%, and 48.34%, respectively. This shows that digital financial inclusion plays the most significant role in the transmission mechanism of mobile payment to consumption upgrading. At the same time, mobile payment can also affect hedonistic consumption through the path of

**Table 9. Conduction mechanism test (II).**

| | (1) | (2) | (3) | (1) | (4) |
|---|---|---|---|---|---|
| | Survival consumption | Development-oriented consumption | Hedonic consumption | Consumption upgrades | Consumption downgrade |
| Digital financial inclusion | 0.1424*** | 0.6609*** | -0.8659*** | 0.0588*** | -0.0451*** |
| | (0.0153) | (0.0289) | (0.0691) | (0.0046) | (0.0045) |
| Mobile payments | 0.1404*** | 0.3497*** | 0.5613*** | 0.0143*** | -0.0166*** |
| | (0.0134) | (0.0253) | (0.0605) | (0.0040) | (0.0040) |
| Control variables | YES | YES | YES | YES | YES |
| Constant terms | -1.9949 | 14.0229*** | -25.9877*** | 2.2466*** | -1.5825*** |
| | (1.7223) | (3.2418) | (7.7639) | (0.5161) | (0.5084) |
| N | 13236 | 13236 | 13236 | 13236 | 13236 |
| R² | 0.4489 | 0.2923 | 0.3053 | 0.0640 | 0.0689 |
| Sobel test | 18.79% | 30.12% | -54.29% | 48.34% | 38.29% |
| F | 539.9640 | 274.3691 | 291.7847 | 46.2699 | 49.9609 |

inclusive digital finance, although its intermediary effect is negative, which we interpret as the "masking effect"; that is, inclusive digital finance obscures the promotion of mobile payment on hedonistic consumption and improves household consumption behavior. In addition, mobile payment inhibits the downgrade of household consumption through inclusive digital finance, exerting an intermediary effect of 38.29%. Therefore, digital financial inclusion plays an important role in the important transmission mechanism of mobile payment affecting consumption categories and consumption upgrading.

## Heterogeneity analysis

1. Analysis of heterogeneity among different income groups

Income level is the main factor determining consumption behavior, and there are significant differences between consumption habits and consumption behaviors of different income groups. There are also certain differences in the consumption impact of mobile payments on households at different income levels. Based on household income levels, this paper divides them into ten levels and adopts categorical regression methods to analyze better the difference in the consumption impact of mobile payment on different income groups. The results show that in addition to 40% to 50% of income groups, mobile payment can effectively promote consumption at all income levels. Among all income groups, mobile payment has the most significant promotion effect on more than 90% and less than 10% of income groups, and in the process of increasing income level, the promotion effect of mobile payment shows a "U" shape. Therefore, mobile payment has the greatest promotion effect on the consumption level of low-income and high-income people, followed by the middle-income group (in Table 10).

2. Heterogeneity of family upbringing structures

The family support structure is an important factor affecting household consumption expenditure. Especially among the young and elderly groups, they are a major source of pressure on family support due to their relatively weak economic status. Generally, the greater the number of children and the elderly, the greater the consumption pressure and the higher the household expenditure. In this paper, we define people under 14 as the adolescent population and those aged 65 and over as the elderly population. On this basis, we divide families into four categories according to the number of children, "childless families," "single-child families," "two-child families," and "multi-child families," and divide families into three categories according to the number of elderly people: "old-age families," "single-old families" and "multi-elderly families," to reflect the family's elderly support pressure. After group regression analysis, Table 11 shows that mobile payment can effectively promote the consumption level of "childless families," "single-child families," "two-child families," and "multi-child families" and has the most significant effect on "multi-child families." In addition, mobile payment has the

**Table 10. Mobile payments and household consumption by income group.**

| Income groups | 10% or less | 10%-20% | 20%-30% | 30%-40% | 40%-50% |
|---|---|---|---|---|---|
| Mobile payments | 0.1809*** | 0.1282*** | 0.0851** | 0.1024** | 0.0621 |
| | (0.0450) | (0.0421) | (0.0411) | (0.0413) | (0.0389) |
| Income groups | 50%-60% | 60%-70% | 70%-80% | 80%-90% | More than 90%. |
| Mobile payments | 0.0783* | 0.0908** | 0.1543*** | 0.1135*** | 0.2148*** |
| | (0.0425) | (0.0361) | (0.0394) | (0.0402) | (0.0561) |

Note: ***, **, * indicate significant at the level of 1%, 5%, and 10%, respectively, and the standard error in parentheses.

**Table 11. Mobile payment and household consumption under different household structures.**

| | Number of child support | | | | Number of elderly dependents | | |
|---|---|---|---|---|---|---|---|
| | **0** | **1** | **2** | **Greater than or equal to 2** | **0** | **1** | **Greater than or equal to 2** |
| Mobile payments | 0.1573*** | 0.1509*** | 0.1393*** | 0.3092*** | 0.1817*** | 0.1647*** | 0.1098*** |
| | (0.0156) | (0.0331) | (0.0535) | (0.1070) | (0.0186) | (0.0273) | (0.0285) |
| Control variables | YES | YES | YES | YES | YES | YES | YES |
| Constant terms | -4.9829** | 4.4968 | -2.2580 | -25.6654* | -3.0764 | -6.4680* | 1.2679 |
| | (1.9924) | (3.9448) | (6.7195) | (13.4660) | (2.1879) | (3.6311) | (4.0588) |
| N | 9772 | 2377 | 875 | 212 | 7661 | 3058 | 2517 |
| R$^2$ | 0.4572 | 0.4201 | 0.3737 | 0.4105 | 0.4350 | 0.4837 | 0.4687 |
| F | 434.2330 | 91.5779 | 28.4474 | 8.7340 | 311.4448 | 151.7267 | 117.8172 |

Note: ***, **, * indicate significant at the level of 1%, 5%, and 10%, respectively, and the standard error in parentheses.

greatest impact on the consumption level of non-elderly households, more significantly than "single-old families" and "multi-old families." In summary, mobile payment can effectively improve the consumption level of families without dependency pressure and families with more dependency pressure.

3. Heterogeneity of family attributes and location

Due to regional differences in mobile payment, household consumption habits, and cultural customs, the impact of mobile payment on household consumption may differ. In order to further study this problem, this paper analyzes the heterogeneity of household distribution from the geographical characteristics of the family distribution. First, the family area is divided into four regions: east, middle, west, and northeast, for analysis, and the specific results are shown in Table 12. It is found that the regression coefficients of mobile payment pass the significance test, which can positively promote the consumption level of households in the eastern, central, western, and northeastern regions. Moreover, mobile payments promote household consumption in the eastern and central regions more than in the western and northeastern regions. This shows that there are certain differences in the consumption impact of mobile payment in different regions, and in the western and northeastern regions, the promotion role of mobile payment still has room for improvement. At the same time, we divided the sample into rural and urban households according to the nature of household registration. We found that mobile payment can positively promote the consumption level of rural and urban households, and the promotion effect on rural households is more significant. This

**Table 12. Mobile payment and household consumption in different regions and urban and rural areas.**

| | **East** | **Middle** | **West** | **Northeast** | **City** | **Countryside** |
|---|---|---|---|---|---|---|
| Mobile payments | 0.1694*** | 0.1775*** | 0.1578*** | 0.1268*** | 0.1395*** | 0.1475*** |
| | (0.0213) | (0.0316) | (0.0249) | (0.0392) | (0.0212) | (0.0174) |
| Control variables | YES | YES | YES | YES | YES | YES |
| Constant terms | -0.8024 | -4.9218 | -3.5997 | 21.8071** | 3.2811 | -9.2283*** |
| | (2.5955) | (5.2319) | (3.9179) | (10.9542) | (2.1820) | (2.5040) |
| N | 4737 | 2601 | 4359 | 1539 | 4821 | 8415 |
| R$^2$ | 0.4523 | 0.4621 | 0.4343 | 0.4086 | 0.3408 | 0.3885 |
| F | 196.5495 | 112.7013 | 168.3101 | 54.1312 | 132.1442 | 282.3698 |

Note: ***, **, * indicate significant at the level of 1%, 5%, and 10%, respectively, and the standard error in parentheses.

**Table 13. Mobile payments, income strata and household consumption.**

| | (1) | (2) | (3) |
|---|---|---|---|
| | Full sample | Rural families | Urban family |
| Mobile payments | 0.2775*** | 0.2385*** | 0.2512*** |
| | (0.0227) | (0.0282) | (0.0462) |
| Income bracket | 0.1457*** | 0.1257*** | 0.1562*** |
| | (0.0068) | (0.0089) | (0.0126) |
| Revenue tiers * Mobile payments | -0.0292*** | -0.0265*** | -0.0232*** |
| | (0.0037) | (0.0051) | (0.0066) |
| Control variables | YES | YES | YES |
| Constant terms | -1.0155 | -7.0558*** | 4.8880** |
| | (1.6529) | (2.4439) | (2.0905) |
| N | 13236 | 8415 | 4821 |
| $R^2$ | 0.4918 | 0.4169 | 0.3963 |
| F | 583.1699 | 287.4580 | 151.6651 |

Note: ***, **, * indicate significant at the level of 1%, 5%, and 10%, respectively, and the standard error in parentheses.

shows that the development and popularization of mobile payment have promoted the consumption vitality of rural households.

## Further analysis

In the previous analysis, we found that mobile payment can effectively promote household consumption among people of different income levels. In order to better compare the differences between different income groups, this paper divides the sample into ten levels according to income level and assigns values of 1–10 as representatives of economic status, among which higher scores indicate higher income levels. In the regression model, we introduce the interaction terms of income hierarchy and mobile payment to analyze and compare the difference in the role of mobile payment in different income brackets. The results show that in Table 13, mobile payment and income level can effectively promote household consumption expenditure. However, the interaction between the two shows a negative correlation, indicating that with the increase in income level, the role of mobile payment on consumption gradually weakens. In addition, after dividing the sample into urban and rural groups, the results were still consistent with the whole sample, and the moderating effect of income level on mobile payment to promote consumption was negatively correlated. In contrast, rural households' moderating effect was greater than urban samples. This means that as household income levels rise, the role of mobile payment in promoting consumption will weaken and weaken.

## Conclusions and policy recommendations

This paper uses data from multiple Chinese household finance surveys to explore the microeconomic performance of mobile payment in household consumption. The empirical results show that mobile payment can effectively improve household consumption. The impact of inclusive digital finance on hedonistic and development-oriented consumption is greater than on survival-oriented consumption, effectively promoting the upgrading of household consumption. Secondly, mobile payment can promote household consumption through digital inclusive financial mechanisms, promote basic and developmental consumption, and reduce the positive impact of mobile payment on hedonistic consumption. Third, the impact of

mobile payment on household consumption varies according to income level, dependency structure, and regional attributes. The promotion role of low-income and high-income groups is more prominent, and the consumption level of low-income groups is effectively improved. In addition, the consumption promotion effect of households in the eastern and central regions is greater than that in the western and northeastern regions, which may be due to the differences in the consumption concepts and habits of households in the eastern and western regions and the difference in the development level of mobile payment. In the impact of income grade on mobile payment to promote consumption, income grade plays a negative moderating role. That is, this promotion effect will decrease with the increase in income level. Finally, the promotion effect of mobile payment in rural households is greater than that of urban households, effectively improving the consumption behavior of rural households.

Based on the above conclusions, this paper puts forward the following policy recommendations:

First, accelerate the popularization and application of mobile payment to boost the consumer market. Build a good interaction mechanism between mobile payment and inclusive digital finance, improve the convenience of consumption and the accessibility of financial services for consumers, and meet the consumption needs of households. In addition, governments can introduce targeted financial services policies to increase digital financial inclusion among poor households, low- and middle-income households, and rural households and narrow the consumption gap.

Second, families in different regions, with different incomes and support structures, borrow mobile technology and big data to provide differentiated digital inclusive financial services and formulate corresponding financial support policies. For example, for low-income families and rural households, more preferential mobile payment policies and services can be introduced; For high-income households, they can be appropriately guided to reduce hedonistic consumption and increase basic and developmental consumption to improve consumption efficiency and quality.

Third, the government can further strengthen mobile payment supervision to protect consumers' legitimate rights and interests. Improve the security mechanism of mobile payment, prevent mobile payment systems from being hacked, strengthen the supervision of mobile payment platforms, and prevent them from suspected fraud, pyramid schemes, and other illegal acts. Strengthen the publicity of digital financial inclusion, and improve consumers' awareness and awareness of digital financial inclusion. Through various media and publicity channels, publicize the benefits and use of digital financial inclusion, improve consumer awareness and trust, and further promote the popularization and use of digital financial inclusion.

Finally, this paper finds that income level negatively moderates the impact of mobile payment on consumption. That is, this promotion effect decreases with the increase in income level. Therefore, the government can take corresponding measures to balance the impact of mobile payment on households at different income levels by adjusting the policy strength and tilt of mobile payment.

## Author Contributions

**Conceptualization:** Ningning Hu.

**Writing – original draft:** Ningning Hu, Guanyu Hou.

**Writing – review & editing:** Ningning Hu, Guanyu Hou.

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
