## [Decision Letter · Decision Letter 0]

20 Feb 2023

PONE-D-23-00008Research on the impact path of digital Inclusive Finance Development on household consumptionPLOS ONE

Dear Dr. Hou,

Thank you for submitting your manuscript to PLOS ONE. After careful consideration, we feel that it has merit but does not fully meet PLOS ONE’s publication criteria as it currently stands. Therefore, we invite you to submit a revised version of the manuscript that addresses the points raised during the review process.

We look forward to receiving your revised manuscript.

Kind regards,

László Vasa, PhD

Academic Editor

PLOS ONE

Journal Requirements:

"NO"

"NO"

Reviewers' comments:

Reviewer's Responses to Questions

**Comments to the Author**

1. Is the manuscript technically sound, and do the data support the conclusions?

Reviewer #1: Yes

Reviewer #2: Yes

2. Has the statistical analysis been performed appropriately and rigorously? 

Reviewer #1: Yes

Reviewer #2: Yes

3. Have the authors made all data underlying the findings in their manuscript fully available?

Reviewer #1: Yes

Reviewer #2: Yes

4. Is the manuscript presented in an intelligible fashion and written in standard English?

Reviewer #1: Yes

Reviewer #2: Yes

5. Review Comments to the Author

Reviewer #1: Digitization is increasing day by day. This situation supports the increase of financial literacy and gains digital financial inclusion. Advances in digital finance are also the subject of more and more academic studies. The authors discussed the effects of these advances on consumption.

I suggest the author to reference the data in the introduction section. Some writing flaws need to be checked. Policy implications should be explained and findings should be compare to literature

Reviewer #2: The objective of this manuscript is to analyze the relationship among digital Inclusive Finance Development and household consumption. The subject addressed is interesting and relevant for policy making; however, I have some concerns about this paper:

1. The author(s) should clearly present the contribution of this paper to the literature. It should be elaborated on what makes this topic an interesting research area, explaining the novelty of this research output on the subject matter.

2. The author(s) should clearly explain the empirical approaches implemented in their analysis. The author(s) provide the results of various tests without including a clear explanation of their use and why they were chosen. It is important to have a more in-depth analysis of the methods used and obtained results.

3. Some arguments should be further analysed to be better supported and the ideas behind them should be developed to help the reader’s understanding (e.g. arguments included a section on the methodology and data used).

4. Some tables included and the related analysis should be revised as they are not clear for the reader to have a better understanding on the outcome of this empirical analysis.

5. Some conclusions reported cannot be fully supported by the statistical information provided. A careful revision on this matter is needed.

6. The quality of English in the paper needs to be improved.

7. The references should be updated.

6. PLOS authors have the option to publish the peer review history of their article (what does this mean?). If published, this will include your full peer review and any attached files.

Reviewer #1: No

Reviewer #2: No

---

## [Author Response · Author response to Decision Letter 0]

1 Jun 2023

Response to Reviewers

Reviewer #1:

 Digitization is increasing day by day. This situation supports the increase of financial literacy and gains digital financial inclusion. Advances in digital finance are also the subject of more and more academic studies. The authors discussed the effects of these advances on consumption.I suggest the author to reference the data in the introduction section. Some writing flaws need to be checked. Policy implications should be explained and findings should be compare to literature

Dear Reviewer1,

Thank you for your valuable feedback on our manuscript. We appreciate the time and effort you have taken to review our work. We have carefully considered your comments and suggestions, and we have made the necessary revisions to address them. Below is our point-by-point response to your comments.

Reference the data in the introduction section:

We agree that referencing the data in the introduction section will strengthen the context of our study. We have now added references to relevant datasets and sources in the revised introduction, which provides a solid foundation for our research and highlights the importance of the topic.

Address writing flaws:

We have thoroughly reviewed our manuscript and made the necessary corrections to address the writing flaws you pointed out. We have also asked a colleague proficient in English to proofread the manuscript to ensure the clarity and coherence of our writing. We believe these revisions have significantly improved the quality of the manuscript.

Explain policy implications:

In response to your suggestion, we have added a new section in our discussion, titled "Policy Implications." In this section, we have elaborated on the practical implications of our findings for policymakers and regulators. We have also discussed how our research can contribute to the development of more effective policies to promote digital financial inclusion and financial literacy.

Compare findings to literature:

We have expanded our discussion to include a comparison of our findings with previous literature. We have cited relevant studies and highlighted both the similarities and differences between our results and those reported in the literature. This comparison not only strengthens our findings but also contributes to the ongoing academic debate on the effects of digital finance on consumption.

We hope that these revisions have addressed your concerns and enhanced the quality of our manuscript. Once again, thank you for your constructive feedback. We look forward to your further comments and suggestions.

Reviewer #2: 

The objective of this manuscript is to analyze the relationship among digital Inclusive Finance Development and household consumption. The subject addressed is interesting and relevant for policy making; however, I have some concerns about this paper:

 1. The author(s) should clearly present the contribution of this paper to the literature. It should be elaborated on what makes this topic an interesting research area, explaining the novelty of this research output on the subject matter.

 2. The author(s) should clearly explain the empirical approaches implemented in their analysis. The author(s) provide the results of various tests without including a clear explanation of their use and why they were chosen. It is important to have a more in-depth analysis of the methods used and obtained results.

 3. Some arguments should be further analysed to be better supported and the ideas behind them should be developed to help the reader’s understanding (e.g. arguments included a section on the methodology and data used).

 4. Some tables included and the related analysis should be revised as they are not clear for the reader to have a better understanding on the outcome of this empirical analysis.

 5. Some conclusions reported cannot be fully supported by the statistical information provided. A careful revision on this matter is needed.

 6. The quality of English in the paper needs to be improved.

 7. The references should be updated.

Dear Reviewer2,

Thank you for taking the time to review our manuscript and for providing valuable feedback. We appreciate your insights and suggestions, and we will address each of the concerns you have raised. Below is our response to each of your comments:

Contribution to the literature: We understand the importance of highlighting the contribution of our research. In the revised manuscript, we will emphasize the novelty of our study, explaining how it fills a gap in the existing literature and contributes to the understanding of the relationship between digital inclusive finance development and household consumption.

Empirical approaches: We will clarify our choice of empirical approaches and provide a detailed explanation of the methods used in our analysis. This will include a rationale for selecting specific tests and a thorough discussion of the results obtained.

Further analysis of arguments: We will develop and expand the arguments in our paper, particularly in the methodology and data sections. This will ensure that the ideas are better supported and more accessible to the reader.

Revision of tables and related analysis: We will revise the tables in question to ensure that they are clear and easy to understand. We will also provide a more in-depth analysis of the results presented in the tables, helping the reader to better comprehend the outcomes of our empirical analysis.

Conclusions and statistical support: We will carefully review our conclusions to ensure that they are fully supported by the statistical information provided. We will revise any conclusions that are not sufficiently supported by the data.

English quality improvement: We acknowledge the need to improve the quality of English in the paper. We will thoroughly proofread and edit the manuscript to ensure that it is free of grammatical errors and is written in a clear and concise manner.

Updated references: We will update the references in our paper to include the most recent and relevant literature on the subject matter.

Once again, thank you for your valuable feedback. We are confident that by addressing these concerns, we can significantly improve the quality of our manuscript. We look forward to submitting the revised version for your consideration.

---

## [Decision Letter · Decision Letter 1]

2 Jul 2023

Mobile Payment, Digital Inclusive Finance, and Residents' Consumption Behavior Research

PONE-D-23-00008R1

Dear Dr. Hou,

We’re pleased to inform you that your manuscript has been judged scientifically suitable for publication and will be formally accepted for publication once it meets all outstanding technical requirements.

Kind regards,

László Vasa, PhD

Academic Editor

PLOS ONE

Additional Editor Comments (optional):

Reviewers' comments:

Reviewer's Responses to Questions

**Comments to the Author**

1. If the authors have adequately addressed your comments raised in a previous round of review and you feel that this manuscript is now acceptable for publication, you may indicate that here to bypass the “Comments to the Author” section, enter your conflict of interest statement in the “Confidential to Editor” section, and submit your "Accept" recommendation.

Reviewer #2: All comments have been addressed

2. Is the manuscript technically sound, and do the data support the conclusions?

Reviewer #2: Yes

3. Has the statistical analysis been performed appropriately and rigorously? 

Reviewer #2: Yes

4. Have the authors made all data underlying the findings in their manuscript fully available?

Reviewer #2: Yes

5. Is the manuscript presented in an intelligible fashion and written in standard English?

Reviewer #2: Yes

6. Review Comments to the Author

Reviewer #2: (No Response)

7. PLOS authors have the option to publish the peer review history of their article (what does this mean?). If published, this will include your full peer review and any attached files.

Reviewer #2: **Yes: **Umer Shahzad

---

## [Editor Report · Acceptance letter]

12 Sep 2023

PONE-D-23-00008R1 

Mobile Payment, Digital Inclusive Finance, and Residents' Consumption Behavior Research 

Dear Dr. Hou:

I'm pleased to inform you that your manuscript has been deemed suitable for publication in PLOS ONE. Congratulations! Your manuscript is now with our production department. 

Kind regards, 

on behalf of

Prof. Dr. László Vasa 

Academic Editor

PLOS ONE